# The role of teams in shaping quality of obstetrical care: a cross-sectional study in Dire Dawa, Ethiopia

Anna D Gage  ,[1,2] Bereket Yakob,[3,4] Margaret McConnell,[1] Tsinuel Girma,[3] Brook Damtachew,[5] Sebastian Bauhoff,[1,6] Margaret E Kruk[1]

[1]Department of Global Health and Population, Harvard University T H Chan School of Public Health, Boston, Massachusetts, USA
[2]Institute for Health Metrics and Evaluation, University of Washington, Seattle, Washington, USA
[3]Fenot Project, The University of British Columbia School of Population and Public Health, Addis Ababa, Ethiopia
[4]Wolaita Sodo University College, Sodo, Ethiopia
[5]Department of Obstetrics and Gynecology, Dil Chora Hospital, Dire Dawa, Ethiopia
[6]Inter-American Development Bank, Washington, District of Columbia, USA

**Correspondence to**
Dr Anna D Gage;
annagage@uw.edu

## ABSTRACT

**Objectives** To examine how characteristics of clinical colleagues influence quality of care.

**Design** We conducted a cross-sectional observational study examining the associations between quality of care and a provider's coworkers, controlling for individual provider's characteristics and contextual factors.

**Setting** Nine health facilities in Dire Dawa Administration, Ethiopia, from December 2020 to February 2021.

**Participants** 824 clients and 95 unique providers were observed across the 9 health facilities.

**Outcome measures** We examine the quality of processes of intrapartum and immediate postpartum care during five phases of the delivery (first examination, first stage of labour, third stage of labour, immediate newborn care and immediate maternal postpartum care).

**Results** For the average client, 50% of the recommended routine clinical actions were completed during the delivery overall, with immediate maternal postpartum care being the least well performed (17% of recommended actions). Multiple healthcare providers were involved in 55% of deliveries. The number of providers contributing to a delivery was unassociated with the quality of care, but a one standard deviation increase in the coworker's performance was associated with a 2% point increase in quality of care (p<0.01); this association was largest among providers in the middle quartiles of performance.

**Conclusions** A provider's typical performance had a modest positive association with quality of delivery care given by their coworker. As delivery care is often provided by multiple healthcare providers, examining the dynamics of how they influence one another can provide important insights for quality improvement.

## STRENGTHS AND LIMITATIONS OF THIS STUDY

⇒ This is among the first studies to examine how multiple providers influence one another to provide quality of care in a low-income healthcare setting.
⇒ We use detailed observations of quality of care with actions attributed to specific providers.
⇒ The study examines cross-sectional associations and the effects cannot be interpreted causally.
⇒ We examine quality of routine deliveries only; further research should examine the role of provider groups in managing complications.

neonatal and maternal care.[3 4] However, in 2018, just 20% of mothers who delivered in a health facility were estimated to receive a high quality of routine maternal care in the Tigray region in northern Ethiopia.[5]

Examinations of poor maternal care quality in similar contexts have found that there is often wide variation in the quality that a single provider will provide across different deliveries. For example, in Uganda the quality of routine actions during labour and delivery varied as much as 60% points across the deliveries for which a single healthcare provider was responsible.[6] In Kenya and Malawi, two studies found that the provider contributed very little to the explained variance in the technical and respectful quality of maternity care.[7 8] The importance of the provider to quality may vary over the course of a delivery, however: another study found that healthcare providers were more important in accounting for the quality of the intrapartum period rather than the assessment or postpartum phases.[9] While these studies have also found that facility, region and country-level factors also play important roles in determining the quality of care, it is worth further considering why provider quality varies in order to shed light on potential improvement strategies.

One limitation of the above studies in their examination of quality is the focus on attribution of quality to a single healthcare provider

## INTRODUCTION

Ethiopia's health system, like those in other low-resource settings, has gaps in maternal and newborn care quality that pose a barrier to improved health outcomes. Although the facility delivery rate has increased dramatically in the past 10 years from 10% in 2011 to 48% in 2019, maternal and newborn mortalities remain high, with 401 maternal deaths per 100 000 live births and 33 neonatal deaths per 1000 live births.[1 2] The majority of such deaths could be avoided with high quality

over the course of a labour and delivery. While many mothers in Ethiopia receive care from a single healthcare provider, this is not the only model. Rather, multiple healthcare providers may attend a single delivery over the course of the labour, delivery and immediate postpartum period, particularly during more complex cases that lead to higher morbidity and mortality. When groups of providers attend a single patient, their interactions with one another may affect the quality of care.[10] Indeed, the disciplines of complexity science and team science suggest that provider groups are more than the collection of independently acting individuals, but rather they may influence one another in non-linear and dynamic ways.[10 11] The culmination of these interactions between providers creates the informal group norms and culture around the quality of care that may affect realised quality and health outcomes.

Groups or teams of providers have been examined frequently in healthcare contexts in high-income countries. For example, studies have found that the quality of HIV care is influenced by the performance of a provider's peers on quality[12] and that a provider's patterns of prescribing medications are affected by a specialist in a provider's network.[13] Similarly, studies have found moderate effects of group characteristics such as professional composition and aspects of teamwork such as coordination on quality of care.[14–16] However, these dynamics have not been examined in healthcare in low-income and middle-income countries. Using cross-sectional observations of delivery care in Ethiopia and adopting a complex systems lens, this study seeks to understand how the number of providers and group dynamics are associated with the quality of delivery care.

## METHODS
### Conceptual framework

We draw on the theory of small groups as complex systems to inform the conceptual framework for this analysis.[11] In this analysis, we define a group of healthcare providers to be all providers who care for a specific woman and newborn over the course of the delivery. Their group identity is thus focused on the shared objective of providing high-quality care, and they draw on one another as well as the broader facility environment in achieving this objective.

Arrow *et al* argue that group behaviour involves interactions at three levels.[11] First, behaviour is influenced by the constituent elements of the groups, in this case the individual providers who bring different experiences, training and roles to the group. Second, there are interactions of the group as an entity, which will include feedback loops between group members. Third, there are interactions with the context in which the group is embedded; this context includes the characteristics of the facility as well as the characteristics of the specific delivery for which the group is providing care.

We focused our analysis on the interactions of the group as an entity, with an emphasis on potential spillovers on quality of care. We also account for care context. Drawing on the peer effects literature, we hypothesise several mechanisms by which group members may influence one another in their provision of quality care. First, members may influence one another through hierarchal relationships defined between providers by their cadre rank or years of experience. For example, a provider working with their superior may perform better due to their better supervision or social pressure. Second, there may be informal influence between providers of different abilities. When working with a high performing group member, there may be positive spillovers because of social pressure, knowledge spillovers or social comparison.[17] Conversely, there may be negative spillovers due to free riding.[17] While our data does not permit investigation into these specific mechanisms, we assessed whether working with a high performing colleague is associated with higher or lower performance.

### Setting

Dire Dawa is a city administration in Eastern Ethiopia with a population in 2019 of approximately 493 000 people, with 63% living in urban Dire Dawa city and the remaining in rural areas surrounding the city.[18] In the 5 years preceding 2019, 84% of pregnant women received at least one antenatal care visit and 69% delivered in a health facility.[18] Despite rising utilisation rates, neonatal mortality increased between 2011 and 2016 to 36 deaths per 1000 live births.[1] In 2021, the administration had 53 public health facilities: 2 hospitals, 15 health centres and 35 health posts. In addition, there were 5 private hospitals and 7 private clinics.

### Data

This analysis is part of a broader study to understand the quality of maternal and immediate postpartum care in Dire Dawa Administration's public health system. Cross-sectional primary data was collected in nine facilities. Public facilities with the highest volumes of deliveries using the 2019 health information system data were included in the study; these facilities were collectively responsible for 83% of the facility deliveries in the region in 2019. The selected facilities consisted of two public hospitals, four health centres in Dire Dawa city and three rural health centres.

Data tools relevant to this analysis include observations of deliveries, interviews with observed clients on discharge from the facility and interviews with all providers who provided care. First, all providers who provided intrapartum or immediate postpartum care in the study facilities were invited to take part in a provider survey that asked about their training, perceptions of the working environment and quality of care and knowledge of complications diagnoses and management.

Second, quality of care was assessed through observations of deliveries by trained health workers. All clients

presenting for delivery during the observation period were invited to participate in the study and their care was observed from the time of arrival at the facility until 6 hours post partum or discharge from the facility. Data collectors identified which items providers completed on a checklist adapted from the Maternal and Child Health Integrated Program tool from the United States Agency for International Development (USAID). For groups of actions (ie, first examination), the collector indicated which provider conducted the actions. For this analysis, only deliveries that were observed from admission to discharge were included, so caesarean section deliveries and intrapartum referrals were excluded. The observation checklist was organised into discrete modules; the ones relevant to this analysis are those for the first examination, checks on the client during the first stage of labour, third stage of labour, immediate newborn care and immediate maternal postpartum care. Finally, all participants whose care was observed were invited to participate in an exit interview on their discharge from the facility.

Observations of care were conducted in most facilities from 22 December 2020 to 20 February 2021. However, until 31 January 2021 one of the hospitals (Sabian Primary Hospital) was not accepting maternity patients because it was a designated COVID-19 treatment facility. Observations of delivery care in Sabian, therefore, occurred between 13 February 2021 and 21 March 2021.

### Outcome

The primary outcome of interest for this analysis was the quality of routine maternal care, adapted from the Quality of Processes of Intrapartum and Immediate Postpartum Care index (QoPIIPC).[19] The original index contains 20 indicators of routine actions that should be conducted during every delivery. To attribute the performance of the indicators to a particular provider, we separated this index into five subindices associated with the module of the checklist (first examination, first stage of labour, third stage of labour, immediate newborn care and immediate maternal postpartum care). For example, the first examination subindex consists of seven items that should be completed during the first examination such as taking pulse and asking whether the client experienced vaginal bleeding. Index items are presented in table 1. Performance on each subindex, which we refer to as the quality of a stage, is calculated as the per cent of actions that were completed so it ranges from 0% to 100%. We attributed the quality of each stage for every delivery to the provider who was noted as providing care during that stage. Deliveries with multiple providers had differences in how stages were assigned, see online supplemental appendix 1 for an example of this structure.

Several of the indicators included a timeliness component, for example, whether the provider checked the mother's vital signs 15 min after birth. To credit providers who completed these actions outside of the designated time frame, we also constructed an alternate index without any time limits on the actions as a sensitivity analysis.

During the first stage of labour 13% of delivery observations did not include the module on checks. We deemed 30% of these missing observations as 'valid missing' if labour was induced directly after the first examination or if there was less than 60 min between the end of the first examination and delivery, as there may not have been time for the enumerator to complete the module. For the remaining observations, non-performance of the actions under the first stage (eg, missing actions imputed as 0) was attributed to the provider who conducted the first examination. Among observations where this stage was not missing, the same provider conducted the first examination and the first stage 65% of the time, while 33% of observations had the same provider for the first and third stages of labour.

### Independent variables

As described above, each stage of a delivery was assigned to a single responsible provider. We defined a group as the providers that cared for a single client at different stages of her delivery. For deliveries with more than one provider, we were interested in how quality for a particular delivery stage completed by the index provider is associated with the characteristics of other providers in the group. While there may be many other providers working at a facility that are unassociated with a delivery, we define these specific provider groups as the index provider's coworkers for the delivery.

We defined three independent variables of interest. First, we examined the cadre of the coworkers relative to the index provider. We created a binary variable for whether the coworkers had a cadre superior to the index provider. There are five categories of cadres, here ranked from low to high: midwife or nurse (diploma), midwife or nurse (Bsc), health officer, general practitioner and integrated emergency surgical officer. Second, we examined the years of professional experience of the coworkers relative to the index provider. Similar to cadre, we created a binary variable for whether coworkers were more experienced than the index provider.

Third, we defined a measure of coworker performance following a two-step approach.[17] In the first step, we created an individual provider performance score for every provider observed, which captures the index provider's own capabilities. We specified a fixed effects model to estimate the provider's ability across all deliveries and delivery stages where they were the responsible provider. We controlled for delivery stage because of the differences in quality between each stage. The provider's fixed effect became a measure of an individual provider's capabilities.

In the second step, we took the average of the provider capabilities measure for all other providers who cared for the same client besides the index provider. As a sensitivity analysis for deliveries with three or more providers, we also compared the average coworker performance with the best coworker's performance. The coworker performance measure was standardised for the analysis.

**Table 1** Components of Quality of Intrapartum and Immediate Postpartum Care Processes index by delivery stage

| | Per cent complete | N |
|---|---|---|
| Average of first examination actions | 47 | 823 |
| Checks woman's HIV status | 68 | 809 |
| Asks whether woman has experienced headaches or blurred vision | 6 | 823 |
| Asks whether woman has experienced vaginal bleeding | 7 | 823 |
| Takes blood pressure during initial client assessment | 73 | 822 |
| Takes pulse during initial client assessment | 67 | 822 |
| Washes hands before initial examination | 13 | 823 |
| Wears gloves before vaginal examination | 99 | 781 |
| Average of first stage of labour actions | 48 | 795 |
| At least once, explains what will happen in labour | 42 | 824 |
| Prepares uterotonic drug to use for AMTSL | 81 | 791 |
| Uses partograph during labour | 53 | 793 |
| Prepares bags and masks for neonatal resuscitation | 14 | 783 |
| Average of third stage of labour actions | 72 | 822 |
| Correctly administers uterotonic | 56 | 820 |
| Assesses completeness of placenta and membranes | 74 | 819 |
| Assesses for perineal and vaginal membranes | 90 | 819 |
| Ties or clamps cord when pulsations stop, or by 2–3 min after birth | 54 | 821 |
| Average of immediate newborn care items | 81 | 694 |
| Immediately dries baby with towel | 98 | 694 |
| Places newborn on mother's abdomen skin-to-skin | 64 | 690 |
| Average of immediate maternal postpartum actions | 17 | 824 |
| Takes mother's vital signs 15 min after birth | 0.2 | 823 |
| Palpates uterus 15 min after birth | 19 | 824 |
| Assists mother to initiate breast feeding | 37 | 695 |
| Overall Quality of Intrapartum and Immediate Postpartum Care Processes index | 51 | 828 |

AMTSL, active management of the third stage of labour.

## Covariates

Drawing on the conceptual framework, we defined several covariates for the individual providers and the contextual environment. First, we included the index provider's cadre and number of years of experience, drawn from the provider interviews. Second, the context environment included characteristics of the client, the delivery and the facility. These include whether the birth had a complication (neonatal resuscitation initiated, newborn referred to the neonatal intensive care unit (NICU), or mother treated for postpartum haemorrhage or eclampsia); whether the delivery was at higher risk for a complication (grand multiparity, mother younger than 18 or older than 35 or multiple births); time of delivery (morning 08:00–18:00 or night 18:00–08:00); and the client's wealth (defined by quintiles within the analytical sample using exit interview asset index). Finally, we also controlled for delivery stage as defined above and facility fixed effects.

## Analysis

We first described the quality of care provided to the study sample on the QoPIIPC index and the characteristics of the clients and providers. We examined whether quality differed by the number of providers caring for the client and examined what factors were associated with group-based care.

We fit a linear mixed-effects model to assess the contribution of group dynamics to quality of care, with observations of delivery stages nested within the index provider. SEs were clustered at the index provider. Covariates were missing for a small number of providers and clients; we used multiple imputation such that all observed deliveries meeting the criteria could be included in the analysis. This analysis was conducted among all group deliveries. In addition, we conducted subanalyses separately for each stage of delivery.

We further investigated how the coworker performance measure interacts with the other group characteristics. First, we categorised the index provider's performance

into quintiles and interacted it with coworker performance to understand how relative performance may be associated with quality. Second, we also ran interaction models between coworker performance and the two measures of provider rank (cadre and years of experience) respectively. We used these interaction models to predict quality across the range of coworker performance holding all other covariates at their means and graphed these marginal models.

## Patient and public involvement

Patients were not involved in the design or implementation of this study.

## RESULTS

Over the data collection period, 983 clients in the 9 facilities were invited to participate in the study. Among them, 979 clients (99.6%) agreed to participate, and 828 clients (84%) were observed throughout the whole delivery and thus met the inclusion criteria for this analysis. The observed clients were cared for by 95 unique providers; 84 (88%) of them were interviewed for the study.

Client and provider characteristics are shown in table 2. In total, 452 (54%) of the observed deliveries took place at one of the two study hospitals, while the remaining deliveries were at the seven health centres. Clients

**Table 2** Delivery and provider characteristics

| | All deliveries | | Hospitals (N=2) | | Health centres (N=7) | |
|---|---|---|---|---|---|---|
| | N | % | N | % | N | % |
| Client and delivery characteristics | | | | | | |
| N deliveries observed | 828 | | 452 | | 376 | |
| Client's primary language | | | | | | |
| Oromiffa | 484 | 63 | 196 | 49 | 288 | 79 |
| Amharic | 178 | 23 | 129 | 32 | 49 | 13 |
| Somali | 81 | 11 | 52 | 13 | 29 | 8 |
| Other | 23 | 3 | 23 | 6 | 0 | 0 |
| Poorest wealth quintile | 150 | 18 | 52 | 12 | 98 | 26 |
| Experienced complication | 213 | 26 | 69 | 15 | 144 | 38 |
| Higher risk pregnancy | 87 | 11 | 45 | 1 | 42 | 11 |
| Time of delivery | | | | | | |
| Day (08:00–18:00) | 341 | 41 | 186 | 41 | 155 | 41 |
| Night (18:00–08:00) | 487 | 59 | 266 | 59 | 221 | 59 |
| Provider characteristics | | | | | | |
| N providers interviewed | 84 | | 50 | | 34 | |
| Years of experience (mean/SD) | 6.2 | 4.5 | 5.9 | 3.6 | 6.5 | 5.7 |
| Female | 54 | 64 | 31 | 62 | 23 | 68 |
| Cadre | | | | | | |
| Midwife or nurse (diploma) | 13 | 15 | 7 | 14 | 6 | 18 |
| Midwife or nurse (BSc) | 55 | 65 | 33 | 66 | 22 | 65 |
| Health officer | 3 | 4 | 0 | 0 | 3 | 9 |
| General practitioner | 7 | 8 | 4 | 8 | 3 | 9 |
| IESO | 6 | 7 | 6 | 12 | 0 | 0 |
| Provider group characteristics | | | | | | |
| Deliveries with >1 provider | 457 | 55 | 326 | 72 | 131 | 35 |
| N providers among group deliveries (mean/SD) | 2.38 | 0.63 | 2.48 | 0.7 | 2.12 | 0.35 |
| At least one health officer, GP or IESO among providers in group | 76 | 16 | 65 | 20 | 11 | 8 |
| At least one provider with over median years of experience | 198 | 43 | 119 | 37 | 79 | 60 |

Delivery complication is neonatal resuscitation, newborn referred to the neonatal intensive care unit or mother treated for postpartum haemorrhage or eclampsia. Higher risk is mother grand multiparous (five or more births), younger than 18 or older than 35 or has multiple births.
GP, general practitioner; IESO, integrated emergency surgical officer .

totalling 213 (26%) experienced a complication during the delivery and 87 (11%) had a higher risk pregnancy. More than one provider had attended 457 (55%) of deliveries over the course of their delivery: 72% of hospital deliveries and 35% of health centre deliveries. Among the deliveries with more than one provider, the mean number of providers was 2.4. Two-thirds of the providers were midwives or nurses (BSc); providers had an average of 6.2 years of cumulative professional experience.

The quality of intrapartum and immediate postpartum care processes provided to the clients in this study was poor (table 1). During the average delivery, only half of the recommended routine actions were done. Actions ranged from 0.2% of mothers whose vitals were checked 15 min after birth to 98% of babies that were dried immediately with a towel. The recommended actions were most often completed for the immediate newborn care stage, while they were least often done during the immediate maternal postpartum care stage. When the index was defined without any time constraints on the actions, average quality of care rose to 58% of recommended actions across the whole delivery (online supplemental appendix 2). However, still only 3% of mothers had their vitals checked after delivery. Over 90% of the variance in quality of care was at the client level, rather than the facility or index provider levels (online supplemental appendix 3).

Between one and five providers cared for a single client and baby over the course of the delivery. The number of providers was unassociated with quality of care (figure 1 and online supplemental appendix 4). The primary determinant of having more than one provider care for a client (online supplemental appendix 5) was the hospital delivery. Time of delivery, client wealth and language were slightly associated with a group delivery, but neither higher risk clients nor complicated deliveries was associated with more providers. The remaining results are only among deliveries with more than one provider.

In the bivariate associations between group characteristics and quality of delivery stages (online supplemental appendix 6), coworkers' performance and seniority by cadre are associated with quality of care. Scatterplots of the index provider and coworker performance (online supplemental appendix 7) further show that this is in part due to quality clustering by facility. Table 3 shows the associations of coworker characteristics with quality of care adjusted for all covariates. After controlling for the facility fixed effect, index provider and client characteristics, a one SD increase in coworker performance was associated with a 2% point increase in quality of care during a given delivery stage or 4% relative to the mean performance. This association was larger during the first stage of labour (4.2% point increase) (online supplemental appendix 8). Working with coworkers that were superior in rank to the index provider was associated with 3.5% point lower quality, while the coworker's relative experience was unassociated with quality. In deliveries with three or more providers, the average coworker performance had a higher association with quality than the maximum coworker performance (online supplemental appendix 9).

Given that each subindex contains between three and seven items, these coefficients translate to less than one additional action completed. However, the coefficients are large in comparison to the risk characteristics: deliveries that had a complication or were from higher risk pregnancies were not more likely to have these routine actions completed than less risky deliveries. The quality of care that women in the wealthiest quintile received was on average 5% points higher than women in the poorest wealth quintile.

The marginal associations of coworker performance by the index provider's capabilities are shown in figure 2. There is no association between coworker performance and quality among the top performing providers: they perform consistently well regardless of their coworkers' performance. However, providers in the middle and low quartiles have large improvement when surrounded by better coworkers. For example, the predicted performance of a third-quartile provider when their coworkers' performance is one standard deviation (SD) above average is 53% in comparison to 48% when their coworkers are one SD below average. Graphs of coworker performance by seniority are included in online supplemental appendices 10 and 11; there are not substantive differences in the associations by either cadre rank or years of experience relative to the index provider.

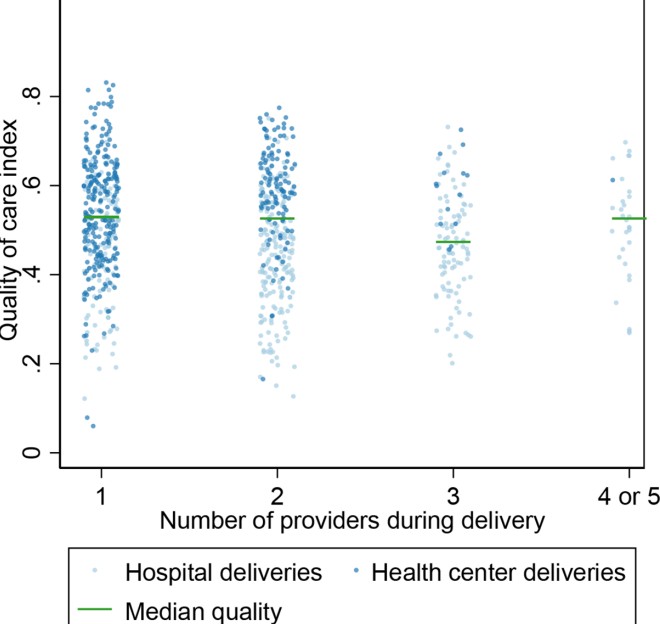

**Figure 1** Quality of intrapartum and immediate postpartum care processes by number of providers and facility type.

## DISCUSSION

Deliveries are often attended by multiple providers who must work together to provide high quality care for

**Table 3** Group dynamics associations with quality of care (outcome range 0–1)

| | Quality of care | | |
|---|---|---|---|
| | Coefficients | P value | 95% CI |
| Group characteristics | | | |
| Peers' performance across all deliveries | 0.023 | 0 | 0.01 to 0.04 |
| Peers are more senior cadre than index | −0.035 | 0.02 | −0.07 to −0.01 |
| Peers are more experienced than index | −0.009 | 0.46 | −0.03 to 0.02 |
| Number of providers | 0 | 0.96 | −0.02 to 0.02 |
| Index provider characteristics | | | |
| Years of experience | −0.002 | 0.41 | −0.01 to 0.00 |
| Provider cadre (midwife or nurse diploma reference) | | | |
| Midwife or nurse Bsc | −0.043 | 0.04 | −0.09 to −0.00 |
| Health officer | −0.141 | 0.02 | −0.25 to −0.03 |
| General practitioner | −0.064 | 0 | −0.11 to −0.02 |
| IESO | −0.066 | 0.48 | −0.25 to 0.12 |
| Context and environment | | | |
| Delivery complication | 0.004 | 0.81 | −0.03 to 0.04 |
| Higher risk pregnancy | 0.003 | 0.88 | −0.04 to 0.04 |
| Night delivery (morning reference) | 0.007 | 0.51 | −0.01 to 0.03 |
| Client wealth index (poorest reference) | | | |
| Wealth 2 | 0.037 | 0.06 | −0.00 to 0.07 |
| Wealth 3 | 0.071 | 0 | 0.03 to 0.11 |
| Wealth 4 | 0.039 | 0.05 | −0.00 to 0.08 |
| Wealth 5 (wealthiest) | 0.05 | 0.01 | 0.01 to 0.09 |
| Delivery stage (first examination reference) | | | |
| First stage of labour | 0.001 | 0.96 | −0.03 to 0.04 |
| Third stage of labour | 0.233 | 0 | 0.20 to 0.26 |
| Immediate newborn care | 0.3 | 0 | 0.26 to 0.34 |
| Immediate maternal postpartum care | −0.32 | 0 | −0.35 to −0.29 |
| Facility (Dil Chorra Hospital reference) | | | |
| Sabien Primary Hospital | 0.017 | 0.35 | −0.02 to 0.05 |
| Biyowale Health Center | 0.127 | 0.01 | 0.04 to 0.22 |
| Legeharae Health Center | 0.129 | 0 | 0.07 to 0.19 |
| Melka Jebdu Health Center | 0.117 | 0 | 0.06 to 0.18 |
| Wahil Health Center | −0.049 | 0.09 | −0.11 to 0.01 |
| Gende Gerada Health Center | 0.083 | 0.05 | 0.00 to 0.16 |
| Goro Health Center | 0.148 | 0 | 0.09 to 0.21 |
| Jelobelina Health Center | 0.16 | 0 | 0.09 to 0.23 |
| Constant | 0.438 | 0 | 0.35 to 0.53 |
| N observed | 2189 | | |

Regressions are at delivery-stage level, with stages nested within providers. Delivery complication is neonatal resuscitation, newborn referred to the neonatal intensive care unit, or mother treated for postpartum haemorrhage or eclampsia. Higher risk is mother grand multiparous (five or more births), younger than 18 or older than 35 or has multiple births.
IESO, integrated emergency surgical officer .

the mother and newborn, particularly in larger health facilities. Adopting a small groups as complex systems framework, this study examined the provider group dynamics and their associations with quality of care in Dire Dawa, Ethiopia. This study had four key findings. First, the observed quality of routine labour and delivery was poor, especially for postpartum maternal care that is vital in timely diagnosis of potentially fatal conditions

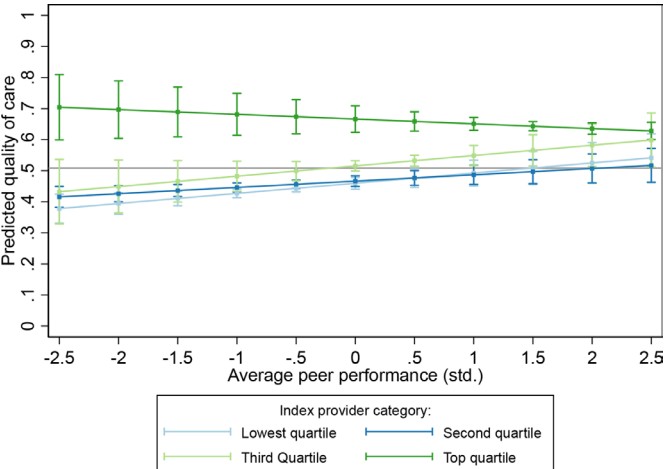

**Figure 2** Predicted quality of care by coworker performance and index provider performance. Points show the predicted estimate of quality by their coworkers' performance for the given level of index provider performance; bars show 95% CIs. Grey line shows median performance.

such as postpartum haemorrhage. Second, women in the poorest wealth quintiles received worse care, even after controlling for facility and provider characteristics. Third, we found that approximately 70% of hospital deliveries and 35% of health centre deliveries had more than one provider involved, but more providers were not associated with higher quality of care. Fourth, providers performed better when working with someone who delivered high quality care, though the differences were small.

We hypothesise several mechanisms by which these provider dynamics may be acting, though our study cannot tease these apart directly. First, we found that if a provider is working with a competent colleague, they do not consequently do less for the client. This is consistent with other peer performance literature, which has shown benefits from working with high-performing colleagues.[12] [17] The association with coworker performance was strongest in the first stage of labour, a stage which may require more communication and coordination between providers to share information after the first examination and before delivery. Greater interaction and interdependence between providers during the delivery may therefore create stronger associations. This aligns with recent work demonstrating that increasing the duration and intensity of collaboration between providers can reduce patient mortality.[20] Contrary to our hypothesis, quality of care was lower when coworkers were of a superior cadre to the index provider. Given that the lower cadres provide the routine actions more often than their senior counterparts, there may be some reverse accountability in working with a more diligent junior colleague. Other work has also shown that junior cadres in Ethiopia have smaller gaps between their knowledge and practice.[21]

This work can inform quality improvement in several ways. First, interventions should place an emphasis on improving immediate postpartum maternal care, which

has the largest identified gaps and the greatest potential for reducing severe maternal morbidity and mortality. While the gaps in monitoring vitals and palpating the uterus that we found in Dire Dawa are larger than those seen in other areas of Ethiopia, the immediate postpartum period has previously been identified as a neglected area.[22] [23] Indeed, in evaluations of quality improvement interventions in Ethiopia, the mother's postpartum care is often not even included as an outcome in favour of immediate newborn care.[24] [25] Interventions may emphasise this area in health worker education or regular supervision. Second, the propensity for multiple providers during delivery suggests that health worker education should emphasise team-based models of care, perhaps through team drills for emergency scenarios. Knowing how to work as a team, assign roles and responsibilities and communicate findings may improve the performance of the team.[26] While our study assessed mainly normal births, highly optimised team-based care will also be required to reduce mortality in the case of obstetrical emergencies.[27]

Third, improvement interventions may consider management or supervision approaches that pair providers of mixed competency levels. Our analysis shows that only the very top performers do not benefit from their coworkers' performance, while most providers may benefit from working with a high performer. Arranging staffing schedules to intentionally pair higher and lower providers may increase the potential for these positive spillovers and strengthen accountability. Supervision approaches typically use traditional cadre hierarchies to define who may act as a supervisor[28]; however, this work suggests that junior cadres may be able to effectively supervise more senior coworkers. Alternately, actual performance could be used to identify potential supervisors.

Finally, the culmination of these interactions between providers over many deliveries creates the facility culture and norms for quality of care. Interventions that aim to change facility norms on quality of care, such as group problem solving or quality improvement collaboratives have shown moderate to high effects on health worker practices in meta-analyses.[29] [30] For example, an improvement intervention in India that used an integrated training, mentoring and a Plan-Do-Study-Act cycle with an emphasis on peer learning improved postpartum monitoring of mothers from 52% to 94%.[31] However, this study also shows the limits of micro-focused interventions. The potential for a 2% point improvement from pairing providers will not overcome the enormous quality deficit seen in labour and delivery care in Dire Dawa. Rather, other macro level strategies such as improved preservice education or redesigning maternal and newborn care service to be provided at facilities with comprehensive emergency obstetric and newborn care will likely be necessary.[32] [33]

This study is among the first to examine how multiple providers influence one another to provide quality of care in a low-income healthcare setting. The detailed observation data with information on who attended each

stage of delivery is a strength of the analysis. There are, however, several limitations to address in future research. First, missing provider attribution for the first stage of delivery required us to make assumptions about who was responsible for the care that was not provided. Second, although we controlled for many potential confounders, the associations may not be interpreted causally. The formation of the provider groups may in some cases be endogenous, for example, if a proactive provider actively seeks support from a high-quality provider to assist with a complex delivery. Third, this study only focused on routine actions that should be done for every mother and newborn. However, the quality of complications management is likely both more impactful for consequent health outcomes and often requires multiple providers to work together coincidentally rather than sequentially. Future research should examine the coordination and communications of providers working to address complications. Finally, this study took place during the COVID-19 pandemic in Ethiopia and for a period during the study, one of the two public hospitals in Dira Dawa was closed to deliveries because it was serving as a COVID-19 treatment facility. This resulted in unusually high delivery volumes in the other hospital which may have altered provider dynamics.

## CONCLUSION

In Dire Dawa, Ethiopia, we find that the number of providers attending to a delivery was not associated with quality of delivery, but the characteristics of those providers in relation to one another did potentially impact quality. As more women across sub-Saharan Africa deliver in hospitals, they are more likely to be attended by multiple healthcare providers, which could have consequences for the quality of care they receive. Unpacking the provider dynamics in how they work together to deliver care quality can yield useful insights for quality improvement in the future.

**Acknowledgements** We gratefully acknowledge the contributions from the data collectors and study participants.

**Contributors** ADG, MK and BY conceived the study. BY, BD and TG oversaw the data collection. ADG conducted the analysis and wrote the first draft with substantial input from all authors. All authors, ADG, BY, MM, SB, BD, TG and MK, critically reviewed the draft and approved this version for submission. ADG and MK are the guarantors of the manuscript and accept full responsibility for the work and conduct of the study, had access to the data, and controlled the decision to publish. The corresponding author attests that all listed authors meet authorship criteria and that no others meeting the criteria have been omitted.

**Funding** We acknowledge funding from the Bill & Melinda Gates Foundation (Grant No. OPP1135922).

**Competing interests** None declared.

**Patient and public involvement** Patients and/or the public were not involved in the design, or conduct, or reporting, or dissemination plans of this research.

**Patient consent for publication** Not applicable.

**Ethics approval** Ethical approval for this study was obtained from Harvard University Institutional Review Board (IRB19-0926), Haramaya University Ethics

Review (IHRERC/138/219) and the Ethiopia National Research Ethics and Review Committee (MoSHE//RD/1411/9403/RO). Participants gave informed consent to participate in the study before taking part.

**Provenance and peer review** Not commissioned; externally peer reviewed.

**Data availability statement** The data are available in a public, open access repository https://dataverse.harvard.edu/dataset.xhtml?persistentId=doi:10.7910/DVN/PWPLME.

**ORCID iD**
Anna D Gage http://orcid.org/0000-0002-4422-0545

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
