## [Reviewer comments · BMJ Open]

ARTICLE DETAILS

TITLE (PROVISIONAL)	The role of teams in shaping quality of obstetric care: a cross-sectional study in Dire Dawa, Ethiopia
AUTHORS	Gage, Anna; Yakob, Bereket; McConnell, Margaret; Girma, Tsinuel; Damtachew, Brook; Bauhoff, Sebastian; Kruk, Margaret

VERSION 1 – REVIEW

REVIEWER	Margareth Portela Fundação Oswaldo Cruz, National School of Public Health
REVIEW RETURNED	13-Aug-2022

GENERAL COMMENTS	It is an interesting manuscript, with clear objectives and hypotheses, appropriate methods, relevant findings and consistent conclusions. I would only recommend that the authors make a text review to correct some minor errors.
--

REVIEWER	Tatyana Rosendo
REVIEW RETURNED	08-Sep-2022

GENERAL COMMENTS	Very interesting study on maternal care quality in a low-income health care setting. Main results: The quality of intrapartum and immediate postpartum care processes provided to the clients in this study was poor and women in the poorest wealth quintiles received worse care, even after controlling for facility and provider characteristics. More providers were not associated with higher quality of care and providers performed better when working with someone who delivered high quality care. The article is well detailed and the STROBE protocol criteria for observational studies were met. The comments below indicate minor adjustments to be made to the text. ABSTRACT: Describe more clearly the main outcome as it's in the method: The primary outcome of interest for this analysis was the quality of routine maternal care, adapted from the quality of processes of intrapartum and immediate postpartum care index (QoPIIPC) it's not in the abstract. METHODS In line 8 of page 7 there is a reference to appendix 1: "We attributed the quality of each stage for every delivery to the provider who was noted as providing care during that stage; see Appendix 1 for an example of this structure". However, Appendix 1 presents "Examples
---

	of provider arrangements during labor and delivery”. On lines 12 to 18 on page 6, the authors report conducting interviews with patients and providers: “Second, we also ran interaction models between coworker performance and the two measures of provider rank (cadre and years of experience) respectively. We used these interaction models to predict quality across the range of coworker performance holding all other covariates at their means and graphed these marginal models.” However, there are no results from these interviews. Could the authors explain why they did not evaluate the interviews? RESULTS: On line 54 on page 11 is written: “Scatterplots of the index provider and coworker performance (Appendix 7) further show that this is in part due to quality clustering by facility.” In fact, appendix 7 presents the table: “Bivariate relationships between independent variables and quality of care”. CONCLUSIONS: The conclusions do not respond to the objectives of the study. I suggest rewriting them.
--	--

VERSION 1 – AUTHOR RESPONSE

Reviewer: 1

Dr. Margareth Portela, Fundação Oswaldo Cruz

Comments to the Author:

It is an interesting manuscript, with clear objectives and hypotheses, appropriate methods, relevant findings and consistent conclusions.

I would only recommend that the authors make a text review to correct some minor errors.

Thank you for reviewing the manuscript. We have reviewed the text and corrected minor errors.

Reviewer: 2

Tatyana Rosendo

Comments to the Author:

Very interesting study on maternal care quality in a low-income health care setting. Main results: The quality of intrapartum and immediate postpartum care processes provided to the clients in this study was poor and women in the poorest wealth quintiles received worse care, even after controlling for facility and provider characteristics. More providers were not associated with higher quality of care and providers performed better when working with someone who delivered high quality care.

The article is well detailed and the STROBE protocol criteria for observational studies were met. The comments below indicate minor adjustments to be made to the text.

Thank you for your careful review and helpful suggestions for improving the manuscript.

ABSTRACT:

Describe more clearly the main outcome as it's in the method: The primary outcome of interest for this analysis was the quality of routine maternal care, adapted from the quality of processes of intrapartum and immediate postpartum care index (QoPIIPC) it's not in the abstract.

We have revised the outcome section of the abstract to emphasize the QoPIIPC measure: "We examine the quality of processes of intrapartum and immediate postpartum care during five phases of the delivery (first exam, first stage of labor, third stage of labor, immediate newborn care, and immediate maternal postpartum care)."

METHODS

In line 8 of page 7 there is a reference to appendix 1: "We attributed the quality of each stage for every delivery to the provider who was noted as providing care during that stage; see Appendix 1 for an example of this structure". However, Appendix 1 presents "Examples of provider arrangements during labor and delivery".

We have revised the text to better represent what is contained in Appendix 1: "Deliveries with multiple providers had differences in how stages were assigned, see Appendix 1 for an example of this structure."

On lines 12 to 18 on page 6, the authors report conducting interviews with patients and providers: "Second, we also ran interaction models between coworker performance and the two measures of provider rank (cadre and years of experience) respectively. We used these interaction models to predict quality across the range of coworker performance holding all other covariates at their means and graphed these marginal models." However, there are no results from these interviews. Could the authors explain why they did not evaluate the interviews?

Table 2 presents some of the key client and provider characteristics that were drawn from these interviews. We chose to only present the characteristics that were most relevant for this analysis; it is outside the scope of the present manuscript to examine all the data collected in those interviews. The Covariates subsection of the Methods has been revised to note that the provider interviews was the source of the provider's cadre and years of experience characteristics.

RESULTS:

On line 54 on page 11 is written: "Scatterplots of the index provider and coworker performance (Appendix 7) further show that this is in part due to quality clustering by facility." In fact, appendix 7 presents the table: "Bivariate relationships between independent variables and quality of care".

Thank you for catching this error, we have corrected the order of the appendix exhibits.

CONCLUSIONS:

The conclusions do not respond to the objectives of the study. I suggest rewriting them.

We have revised the conclusions section to better reflect our findings corresponding to the study objectives.